# DRAFTATTENTION: FAST VIDEO DIFFUSION VIA LOW-RESOLUTION ATTENTION GUIDANCE

## ABSTRACT

Video generation models based on diffusion transformers have recently attracted widespread attention for their excellent generation quality. Despite recent progress, their computational expense remains the principal bottleneck. In particular, attention alone accounts for more than 80% of the overall latency, and the synthesis of only 8 seconds 720p video takes tens of minutes, which severely restricts practical applicability and scalability. To address this, we propose **DraftAttention**, a training-free framework for the acceleration of video diffusion transformers with dynamic sparse attention on GPUs. The key idea is to compute the low-resolution draft attention based on the downsampled low-resolution query and key with minor computational overhead. The draft attention exposes redundancy both spatially within each feature map and temporally across frames, thus identifying the most important areas in the attention map. The resulting low-resolution sparse mask then guides full-resolution sparse attention computations. To align region-level sparsity with token-level computations, we further propose a deterministic reordering of tokens such that entries in each region become contiguous in memory, ensuring hardware-friendly execution of sparse attention. Our theoretical analysis demonstrates that the low-resolution draft attention closely approximates the full attention, providing reliable guidance for constructing accurate sparse attention. Experimental results show that our method outperforms existing sparse attention approaches in video generation quality and achieves up to $2\times$ end-to-end speedup on GPUs.

## 1 INTRODUCTION

Diffusion Transformers (DiTs) (Peebles & Xie, 2022) have emerged as a powerful paradigm for visual generative tasks across both image and video generation, surpassing the traditional UNets (Ronneberger et al., 2015). Video generation with DiTs adopts spatiotemporal 3D full attention to extend image-based generation to the temporal domain (Arnab et al., 2021), leading to visually coherent high-quality video generation performance (Yang et al., 2024; Kong et al., 2024; Wang et al., 2025), validating the effectiveness of DiTs for video generation. Despite the superior generation performance with DiTs, it remains computationally expensive due to the attention mechanism in transformers.

The quadratic complexity with respect to context length (Dao et al., 2022) becomes a significant computational bottleneck when handling sequences with hundreds of thousands of tokens. For example, as shown in Figure 1, the Hunyuan Video model (Kong et al., 2024) spends over 80% of its total computation on the attention mechanism when generating videos longer than 16 seconds. As a result, the slow generation speed limits the application and deployment of these promising video generation models across a range of practical tasks.

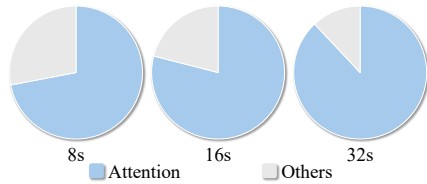

Figure 1: FLOPs breakdown for 720p video generation with Hunyuan Video.

Fortunately, pioneering works (Zhang et al., 2023; Tang et al., 2024; Xiao et al., 2023; Jiang et al., 2024) on Large Language Models (LLMs) (Radford et al., 2019; Touvron et al., 2023a;b; Grattafiori et al., 2024) have demonstrated substantial redundancy in the attention mechanism, offering an opportunity for acceleration by introducing sparsity into the attention. Inspired by this, recent works (Xi et al., 2025; Xia et al., 2025) explore the sparse attention methods for video generation models, demonstrating promising speedups while preserving generation quality. Specifically, two

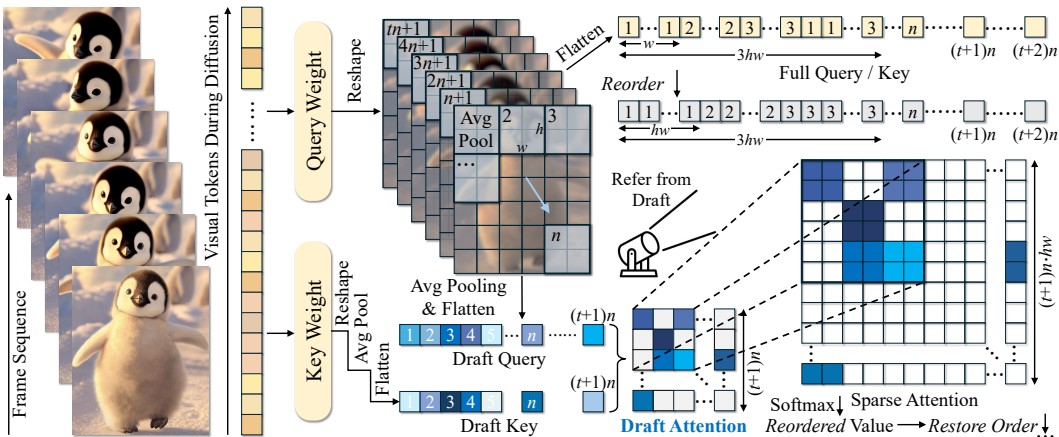

Figure 2: Whole **DraftAttention** Pipeline. Both the query and key are reshaped into sequences of feature maps across frames, then downsampled via average pooling to produce the low-resolution draft query and draft key. Draft attention is computed using the flattened draft query and key. The full-resolution query and key need to be reordered for the alignment of draft attention guidance.

static sparse attention patterns (targeting spatial and temporal dimensions respectively) are explored in Sparse VideoGen (Xi et al., 2025) to reduce redundancy, with relatively significant performance degradation under large sparsity because of non-adaptive static patterns. To mitigate this issue, dynamic sparse attention is investigated in AdaSpa (Xia et al., 2025) to perform full attention once for different prompts as a warm-up to guide subsequent sparsity. Although AdaSpa provides prompt-dependent sparse patterns, sparse attention still remains static during the diffusion process.

**Novel Framework.** Motivated by the absence of true dynamic sparse attention at the per-module level, we investigate a more fine-grained design—adapting the sparse attention patterns dynamically for each specific attention module. In this paper, we propose an efficient sparse attention method, **DraftAttention**, as shown in Figure 2, which leverages draft attention to dynamically generate a sparse pattern for each attention module, enabling efficient acceleration of video diffusion transformers. The key idea is to compute the draft attention based on downsampled low-resolution query and key, thus identifying the most important areas in the attention map with minor computational overhead. The resulting low-resolution sparse mask then guides full-resolution sparse attention, with effective reordering applied to ensure fast, hardware-friendly execution.

**Superior Advantages.** We highlight the following advantages with our draft attention method: **(i)** (Efficiency) The computation of draft attention map is lightweight, as it operates on a reduced number of tokens, thereby lowering the quadratic complexity of the attention mechanism. **(ii)** (Effectiveness) The draft attention captures high-level representations and preserves essential visual patterns for videos, leading to an effective mask to identify the critical structures in attention mechanism. **(iii)** (Plug-and-Play) Our method requires no additional training and integrates seamlessly as a plug-and-play module into existing video diffusion transformers for handling long input sequences.

**Theoretical Justification.** We further present the theoretical analysis that formally characterizes how the low-resolution draft attention effectively guides the full-resolution attention mechanism. Specifically, we show that the upper bound of the difference between the full-resolution attention map and the draft attention map remains controlled. Meanwhile, we show that the error introduced by the sparse pattern derived from the draft attention map remains bounded.

**Hardware Friendliness.** To align the region-level sparsity with token-level computations, we apply a deterministic reordering of tokens such that entries in each region become contiguous in memory, ensuring hardware-friendly execution of sparse attention. Specifically, through reordering, we group the scattered sparse patterns into a contiguous format, allowing all visual tokens within each kernel to be processed in a single stage—either computed or skipped. This enables both accurate and faster sparse attention at full resolution.

**Comprehensive Evaluation.** In our experiments, we use an $8 \times 16$ pooling kernel with a stride equal to the kernel size, reducing the number of tokens by a factor of 128. This configuration also matches the efficient block size supported by efficient attention computation frameworks (Dao et al.,

2022; Guo et al., 2024). Such aggressive downsampling also incurs minimal computational overhead for the low-resolution draft attention. Our comprehensive evaluation demonstrates that our method outperforms other sparse attention methods on video generation tasks across various resolutions under the same computational budget. It achieves up to a $2\times$ end-to-end speedup on GPUs, demonstrating strong practical efficiency and scalability for long video sequences without compromising generation quality. Our contributions are summarized as follows,

**1.** We introduce a vision-centric perspective on spatial and temporal redundancy in video diffusion, using pooling to extract high-level representations with a broader receptive field. Building on this, we propose **DraftAttention**, a hardware-friendly approach that accelerates video diffusion transformers using guidance from low-resolution draft attention.

**2.** We provide a theoretical analysis demonstrating the controlled difference between full-resolution attention and low-resolution draft attention, as well as the bounded error introduced by the sparse pattern derived from the draft attention map, thereby justifying the effectiveness of our design.

**3.** Experimental results show that DraftAttention achieves better video generation quality compared to other sparse attention methods with same computation cost. Meanwhile, on GPUs, our method achieves up to $2\times$ end-to-end acceleration for video generation.

## 2 RELATED WORKS

### 2.1 EFFICIENT DIFFUSION MODELS

**Diffusion Model Compression.** Weight quantization is a common approach to compress diffusion models and achieve acceleration (Li et al., 2023a). Previous works (Zhang et al., 2025b;a; Li* et al., 2025) propose optimal quantization methods to quantize attention weights to INT8, INT4/FP8, or even FP4, which achieve high compression ratios for the diffusion model size. Moreover, other works explore efficient architectures (Xie et al., 2025) including linear attention or high-compression auto-encoders (Chen et al., 2025) to accelerate the diffusion and improve model performance, which improves the scalability of diffusion models. Our method is orthogonal to these techniques and can be integrated with them to yield additional performance gains.

**Reduce Diffusion Steps.** Some distillation-based works (Li et al., 2023b; Yin et al., 2024) adopt training with distillation to build few-step diffusion models, which accelerates the diffusion progress by reducing the steps. However, such distillation techniques require expensive re-training or fine-tuning, which is impractical for the application of most video diffusion models. In contrast, our approach directly uses off-the-shelf pre-trained models without any additional training.

### 2.2 SPARSE ATTENTION METHODS

Attention mechanisms exhibit inherent sparsity (Child et al., 2019), allowing computational acceleration by limiting interactions to a subset of the key-value pair. StreamingLLM (Xiao et al., 2023) explores the temporal locality with attention sinks to further preserve sparse attention model performance. H2O (Zhang et al., 2023) identifies a small set of Heavy Hitter tokens that dominate overall attention scores. DuoAttention (Xiao et al., 2025) and MInference (Jiang et al., 2024) demonstrate distinct sparse patterns across different attention heads. XAttention (Xu et al., 2025) leverages the sum of antidiagonal values in the attention matrix to provide a powerful proxy for block importance, resulting in high sparsity and dramatically accelerated inference. Sparse VideoGen (Xi et al., 2025) explores spatial and temporal heads in video diffusion models to improve the inference efficiency. AdaSpa (Xia et al., 2025) applies dynamic block-sparse masking with online token importance search, accelerating video diffusion without fine-tuning. These works collectively show that such transformer-based models contain significant redundancy in their attention mechanisms. This motivates our exploration of dynamic, fine-grained sparse attention patterns for video diffusion transformers.

## 3 METHODOLOGY

We introduce the framework of our draft attention in great detail to first identify critical areas in draft attention with a low-resolution mask and then apply the mask to full-resolution attention. Next theoretical analysis for the draft attention and the corresponding sparse attention is presented to

demonstrate the effectiveness of our design. Moreover, we provide a deterministic reordering of tokens to align the region-level sparsity with token-level computation, ensuring efficient hardware-friendly execution.

## 3.1 DRAFT ATTENTION

Full attention over long video sequences is prohibitively expensive due to its quadratic complexity in sequence length. However, many interactions in video are spatially and temporally localized. We leverage this structure by introducing a two-stage attention mechanism: a lightweight *draft attention* phase that estimates regional relevance, followed by a masked sparse attention applied to the full-resolution sequence. We first define the full attention computation below.

**Definition 3.1** (Full Attention). *Given hidden states $X \in \mathbb{R}^{n \times d}$, the full attention output is:*

$$\mathsf{Attn}(X) = \mathsf{Softmax}\left(\frac{QK^\top}{\sqrt{d}}\right) V \in \mathbb{R}^{n \times d}, \tag{1}$$

*where $Q = XW_Q$, $K = XW_K$, $V = XW_V$ are the query, key, and value projections, and $W_Q, W_K, W_V \in \mathbb{R}^{d \times d}$ are learned weight matrices.*

To reduce computation, we downsample $Q$ and $K$ via average pooling, forming a low-resolution draft attention map to guide sparsity.

**Definition 3.2** (Draft Attention via Average Pooling). *Given hidden states $X \in \mathbb{R}^{n \times d}$, representing spatial-temporal tokens across frames, we partition the sequence into $g \ll n$ disjoint regions $\{R_i\}_{i=1}^g$, where each region $R_i \subset [n]$ corresponds to a pooled spatial patch over time. Each $R_i$ is an unordered set of token indices. Let $Q$ and $K$ be the projected queries and keys. The draft query and draft key representations are obtained by average pooling over each region:*

$$\widetilde{Q}_i = \frac{1}{|R_i|} \sum_{j \in R_i} Q_j, \quad \widetilde{K}_i = \frac{1}{|R_i|} \sum_{j \in R_i} K_j, \quad for\ i = 1, \dots, g. \tag{2}$$

*The resulting low-resolution **draft attention map** is computed as:*

$$A_{\mathrm{draft}} = \mathsf{Softmax}\left(\frac{\widetilde{Q}\widetilde{K}^\top}{\sqrt{d}}\right) \in \mathbb{R}^{g \times g}. \tag{3}$$

*This map approximates region-level relevance and is used to guide sparse attention over the full-resolution sequence.*

The computation cost of the low-resolution draft attention map is minor compared with the full-resolution attention computation, as it operates on a reduced number of tokens and thereby lowers the quadratic complexity of the attention mechanism.

**Guided Sparsity via Draft Attention.** To reduce the cost of full attention, we extract a structured sparsity pattern from the draft attention map $A_{\mathrm{draft}} \in \mathbb{R}^{g \times g}$ by retaining only a fraction $r \in (0, 1)$ of the most salient region-to-region interactions. We define a binary *indicator* mask $M \in \{0, 1\}^{g \times g}$, where $M_{ij} = 1$ indicates that region $R_i$ is permitted to attend to region $R_j$, and $M_{ij} = 0$ otherwise. The mask is constructed by selecting the top-scoring entries in $A_{\mathrm{draft}}$ under a fixed sparsity ratio $r$.

To align the region-level sparsity with token-level computation, we apply a deterministic reordering of tokens such that entries in each region $R_i$ become contiguous. This facilitates efficient masking and block-wise computation in sparse attention. We provide more details for reordering in Section 3.3.

This region-level sparsity pattern is then lifted to token resolution by defining a full-resolution attention mask $\widehat{M} \in \{0, -\infty\}^{n \times n}$:

$$\widehat{M}_{uv} = \begin{cases} 0, & \text{if } M_{ij} = 1 \text{ with } u \in R_i,\ v \in R_j, \\ -\infty, & \text{if } M_{ij} = 0 \text{ with } u \in R_i,\ v \in R_j. \end{cases} \tag{4}$$

In general, the attention map is split into multiple non-overlapping regions by the pooling kernels. For each region, all its elements are either computed for attention or skipped for acceleration. The

determination for whether to skip each region is denoted by the low-resolution binary indicator mask $M$ for all regions, with $\widehat{M}$ as its full-resolution mask for all elements (i.e., tokens).

Sparse attention is then computed by applying the additive mask before the softmax:

$$\mathsf{SparseAttn}(X) = \mathsf{Softmax}\left(\frac{QK^\top}{\sqrt{d}} + \widehat{M}\right)V. \tag{5}$$

This formulation retains the most relevant interactions while enforcing structured sparsity for improved computational efficiency.

## 3.2 THEORETICAL ANALYSIS

We present Frobenius-norm bounds quantifying the error introduced by our two-stage approximation strategy: (1) average pooling (draft attention), and (2) structured sparsification via top-$r$ indexing.

### 3.2.1 ERROR FROM DRAFT ATTENTION

Let the input sequence be partitioned into $g$ disjoint regions $\{R_i\}_{i=1}^g$ of equal size $|R_i| = n/g$. Define the full-resolution attention logits and their pooled approximation as:

$$S_{uv} := \langle Q_u, K_v \rangle, \quad \widetilde{S}_{ij} := \langle \widetilde{Q}_i, \widetilde{K}_j \rangle, \qquad u, v \in [n],\ i, j \in [g], \tag{6}$$

where $\widetilde{Q}_i = \frac{1}{|R_i|}\sum_{u \in R_i} Q_u$ and similarly for $\widetilde{K}_j$. We restore the region-level scores $\widetilde{S} \in \mathbb{R}^{g \times g}$ to full resolution by defining a block-constant approximation:

$$(S_{\text{draft}})_{uv} := \widetilde{S}_{ij} \quad \text{for } u \in R_i,\ v \in R_j. \tag{7}$$

Define the worst-case deviation between token-level logits and their region-averaged counterpart as:

$$\delta := \max_{i,j}\ \max_{u \in R_i,\, v \in R_j}\ \left| S_{uv} - \widetilde{S}_{ij} \right|. \tag{8}$$

**Theorem 3.3** (Draft Attention Error). *If all regions have equal size $|R_i| = n/g$, then the Frobenius-norm error between the full and draft logit matrices is bounded by:*

$$\|S - S_{\text{draft}}\|_F \leq \delta\, n. \tag{9}$$

The detailed proof of Theorem 3.3 is shown in Appendix A.

**Remark 3.4.** *Theorem 3.3 quantifies the approximation error introduced by replacing token-level attention logits with block-wise averages obtained via average pooling. In practice, if tokens within a region are similar—such as in videos with local temporal consistency or spatial smoothness—the difference $|S_{uv} - \widetilde{S}_{ij}|$ remains small for most $(u, v)$. Consequently, the overall Frobenius-norm error $\|S - S_{\text{draft}}\|_F$ scales with a modest $\delta$, leading to minimal distortion in the attention structure. This justifies using the low-resolution draft map as a proxy for full-resolution attention in computationally constrained settings.*

### 3.2.2 ERROR FROM SPARSITY MASK

We now consider the additional error introduced by sparsifying attention via a top-$r$ region selection guided by the draft scores. Let $\widetilde{S}_{(1)} \geq \cdots \geq \widetilde{S}_{(g^2)}$ be the sorted region-level scores and define the threshold $t := \widetilde{S}_{(\lceil rg^2 \rceil)}$. Let $M_{ij} = 1$ if $\widetilde{S}_{ij} \geq t$ and 0 otherwise. Lifting this region mask to token resolution yields the mask $\widehat{M} \in \{0, -\infty\}^{n \times n}$ in Equation (4).

**Theorem 3.5** (Sparsity Mask Error). *Let $S = QK^\top/\sqrt{d}$, $P = \mathsf{Softmax}(S)$, and $P^{\widehat{M}} = \mathsf{Softmax}(S + \widehat{M})$ where $\widehat{M} \in \{0, -\infty\}^{n \times n}$ is the attention mask induced by top-$r$ draft selection. Assume uniform regions and intra–region deviation at most $\delta$ as in Eq. (8). If each row retains at least a fraction $s$ of tokens under the thresholding rule, then*

$$\|P - P^{\widehat{M}}\|_F \ \leq\ \frac{2\sqrt{n}}{1 + \frac{s}{1-s}\, e^{-2\delta}}. \tag{10}$$

Figure 3: Illustration for the necessity of the reordering. The "x$y$" in attention map denotes attentivity between token x in query and token $y$ in key. Grouping the sparse pattern enables hardware-friendly layout, leading to faster attention computation.

The detailed proof of Theorem 3.5 is shown in Appendix A.

**Remark 3.6.** *The bound in Equation* (10) *highlights two controlling factors: the intra–region deviation $\delta$ and the minimal keep ratio $s$ determined by the draft threshold. A smaller $\delta$ means tokens within each region are more homogeneous, so pruning introduces little distortion. A larger $s$ means fewer entries are removed, so the dropped probability mass is small. In practice, this means draft attention is most reliable when the regional structure is coherent and the pruning threshold is not overly aggressive, ensuring that sparse attention closely matches the dense baseline.*

Together, Theorems 3.3 and 3.5 provide a principled decomposition of the total approximation error: one from average pooling, and one from sparsity. Their combined bound shows that draft attention is an efficient surrogate for full attention, maintaining structural fidelity while enabling substantial computational savings. This justifies its use in long-context video diffusion transformers, where local smoothness and sparse relevance patterns are common.

## 3.3 REORDERING FOR PATCH-ALIGNED SPARSE ATTENTION

To enable accurate and efficient sparse attention that respects spatial structure, we apply a deterministic reordering algorithm (Algorithm 1) to the flattened full-resolution token sequence. As shown in Figure 3, the goal is to align the memory layout of full-resolution tokens with the spatial region structure used in low-resolution draft attention. This alignment ensures that the region-level sparsity patterns are directly and efficiently propagated to full-resolution attention through block-level masking.

**Justification.** In the default row-major layout, spatial tokens are appended row-wise within each frame, causing spatial patches to be scattered in memory. This fragmentation hinders efficient usage of sparse attention kernels, which rely on contiguous blocks in fixed size for the optimal performance. As illustrated in Figure 3, tokens 1, 2, 5, and 6 are spatial neighbors but are not stored consecutively in the memory of full attention map (i.e., left side of Figure 3) due to the presence of tokens 3 and 4. While it is still possible to gather these tokens and compute their average, this process is highly inefficient. Similarly, masking out these scattered blocks is also inefficient, as it reduces the block size, which in turn lowers arithmetic intensity, causes uncoalesced memory access, and increases the number of kernel launches.

---

**Algorithm 1:** Generate Reorder Index

**Input:** Frame size $(H, W)$, patch size $(h, w)$, number of frames $F$
**Output:** Permutation $\pi \in [n]$ where $n = F \cdot H \cdot W$

$\pi \leftarrow []$;
**for** $f = 0$ **to** $F - 1$ **do**
  **for** $i = 0$ **to** $H/h - 1$ **do**
    **for** $j = 0$ **to** $W/w - 1$ **do**
      **for** $u = 0$ **to** $h - 1$ **do**
        **for** $v = 0$ **to** $w - 1$ **do**
          $y \leftarrow i \cdot h + u, x \leftarrow j \cdot w + v$;
          `idx` $\leftarrow f \cdot H \cdot W + y \cdot W + x$;
          Append `idx` to $\pi$;

**return** $\pi$

---

**Design.** We divide each frame into non-overlapping patches of size $h \times w$. For each frame, tokens within the same patch are grouped contiguously. Unlike prior methods (e.g., SVG (Xi et al., 2025)) that overlook misalignment issues when the kernel size does not divide evenly into the latent feature map size, our per-frame design preserves the completeness of each feature map, generating more reliable captured high-level representations. Meanwhile, this per-frame design ensures that each patch in a frame is stored as a contiguous block, matching the structure of the downsampled low-resolution

queries and keys used in draft attention. For instance, tokens 1, 2, 5, and 6 belong to the same patch and are reordered to appear consecutively in both the query and key sequences, as illustrated at the top of Figure 3. Reordering ensures that each entry in draft attention map (e.g., a$a$) corresponds to a specific block ($\{1, 2, 5, 6\}$ from query and $\{1, 2, 5, 6\}$ from key) within reordered full attention map.

**Execution.** Applying the permutation $\pi$ ensures that tokens grouped in each $h \times w$ patch are stored contiguously in memory, enabling efficient block-wise indexing and masking. This structured layout aligns the memory access pattern with the computational needs of sparse attention operations. This is especially critical for efficient execution with frameworks like FlashAttention (Dao et al., 2022) , which leverage fused GPU kernels that operate on fixed-size blocks.

**Restoration.** After sparse attention is applied in the reordered space (i.e., the attention computation for reordered query, key, and value), we apply the inverse permutation $\pi^{-1}$ (Algorithm 2) to restore the original spatial-temporal layout for the following correct model inference.

**Benefit.** This reordering bridges the gap between the coarse-grained sparsity structure derived from draft attention and the fine-grained full-resolution attention computation. This layout guarantees that pooled regions align cleanly with memory blocks, preserving spatial locality and enabling predictable, coalesced memory access. As a result, it supports efficient masking and ensures compatibility with high-throughput attention kernels. This design significantly reduces overhead and maximizes hardware efficiency during attention computation.

---

**Algorithm 2:** Generate Restore Index

**Input:** Permutation $\pi \in [n]$
**Output:** Inverse permutation $\pi^{-1}$
Initialize $\pi^{-1} \leftarrow$ zero array of length $n$;
**for** $i = 0$ **to** $n - 1$ **do**
    $\pi^{-1}_{\pi_i} \leftarrow i$;
**return** $\pi^{-1}$

---

## 4 EXPERIMENTAL RESULTS

### 4.1 EXPERIMENT SETUP

**Model Family.** We adopt open-sourced state-of-the-art video generation models in our experiments, including HunyuanVideo-T2V (Kong et al., 2024) for 768p resolution with 128 frames and Wan2.1-T2V (Wang et al., 2025) for both 512p and 768p resolutions with 80 frames. We use 512p and 768p resolutions to align with the 8×16 average pooling kernel (with stride equal to the kernel size), enabling convenient and consistent downsampling of visual tokens during the diffusion process. This is because the corresponding latent sizes—32×48 for 512p and 48×80 for 768p—are perfectly divisible by the 8×16 kernel, ensuring efficient and artifact-free pooling. Note that our method supports video generation at any resolution by applying appropriate padding. Following prior works (Xi et al., 2025; Li et al., 2024a;b; Liu et al., 2024), we retain full attention across all methods for the first 25% of denoising steps to preserve the video generation quality. We adopt Block Sparse Attention (Guo et al., 2024) for the implementation of our method and mainly compare our method with the Sparse VideoGen (SVG) (Xi et al., 2025). We observe discrepancies in the generation results of the Wan2.1-T2V model between our method and SVG, due to difference of codebases. To ensure a fair comparison, we provide results using full attention for both methods.

**Metrics and Prompts.** We evaluate the quality of generated videos with VBench (Huang et al., 2024), and the similarity of generated videos with metrics including Peak Signal-to-Noise Ratio (PSNR), Structural Similarity Index Measure (SSIM), and Learned Perceptual Image Patch Similarity (LPIPS) (Zhang et al., 2018). Especially, we report the image quality, subject consistency, background consistency, dynamic degree, and aesthetic quality from VBench for our generated videos. All videos are generated with prompts from the Penguin Video Benchmark (Kong et al., 2024) by HunyuanVideo. The reported computation cost in PFLOPs includes the main diffusion transformer models, and the latency results are all tested on H100 and A100. More details are shown in Appendix B.

### 4.2 MAIN RESULTS

**Higher Generation Quality.** We provide the main results compared with the SVG method in Table 1. To perform a comprehensive study, different sparsity ratios for the attention mechanism are evaluated

Table 1: Main results of our method compared to the Sparse VideoGen (SVG) Xi et al. (2025).

| Model | Method | Sparse Ratio | PSNR ↑ | SSIM ↑ | LPIPS ↓ | Img. Qual. | Sub. Cons. | Bakg. Cons. | Dyn. Deg. | Aes. Qual. | PFLOPs ↓ |
|---|---|---|---|---|---|---|---|---|---|---|---|
| Wan2.1 (512p) | SVG | 0% | / | / | / | 65.1% | 95.0% | 95.9% | 44.7% | 58.9% | 145.65 |
| | | 55% | 25.61 | 83.63 | 10.42 | 65.2% | 94.8% | 95.9% | 45.2% | 58.9% | 99.26 |
| | | 75% | 23.66 | 78.80 | 15.05 | 64.7% | 94.5% | 95.7% | 45.7% | 58.6% | 91.12 |
| | Ours | 0% | / | / | / | 69.3% | 95.5% | 96.7% | 47.6% | 61.5% | 145.65 |
| | | 55% | 25.13 | **84.77** | **8.43** | 69.2% | 95.5% | 96.6% | 47.6% | 61.5% | 99.26 |
| | | 75% | 23.10 | **79.07** | **12.37** | 69.0% | 95.4% | 96.5% | 46.9% | 61.5% | 91.12 |
| Wan2.1 (768p) | SVG | 0% | / | / | / | 67.7% | 95.3% | 96.4% | 43.4% | 60.4% | 609.52 |
| | | 55% | 26.01 | 84.81 | 10.89 | 67.9% | 95.1% | 96.3% | 42.1% | 60.0% | 354.68 |
| | | 75% | 23.62 | 79.05 | 17.57 | 67.5% | 94.8% | 96.1% | 42.1% | 58.8% | 309.95 |
| | Ours | 0% | / | / | / | 67.5% | 95.7% | 97.1% | 37.7% | 60.8% | 609.52 |
| | | 55% | **29.22** | **92.16** | **5.82** | 67.4% | 95.6% | 97.0% | 37.2% | 60.8% | 354.69 |
| | | 75% | **27.17** | **88.97** | **8.71** | 67.2% | 95.6% | 97.0% | 38.6% | 60.7% | 309.95 |
| Hunyuan (768p) | Dense | 0% | / | / | / | 66.4% | 96.0% | 97.0% | 36.4% | 58.6% | 682.67 |
| | SVG | 60% | 25.80 | 84.46 | 14.20 | 66.4% | 95.9% | 97.0% | 36.6% | 58.2% | 343.72 |
| | | 80% | 24.70 | 81.90 | 17.55 | 66.0% | 95.7% | 96.9% | 33.9% | 58.1% | 295.30 |
| | | 90% | 23.48 | 78.57 | 22.60 | 65.1% | 95.4% | 96.7% | 32.8% | 57.5% | 283.20 |
| | Ours | 60% | **32.08** | **93.21** | **5.58** | **66.4%** | **95.9%** | **97.0%** | 35.9% | **58.5%** | 343.73 |
| | | 80% | **29.19** | **89.32** | **9.19** | **66.2%** | **95.8%** | **97.0%** | 35.7% | **58.2%** | 295.31 |
| | | 90% | **24.22** | **79.90** | **18.12** | **65.9%** | **95.7%** | **96.9%** | 36.6% | **57.8%** | 283.20 |

Figure 4: Latency results tested in 768p with H100 and A100 for different sparsity ratios in attention.

under various resolutions with multiple video generation model architectures. With the Wan2.1 model, we observe that our method achieves less image quality degradation compared with SVG. The similarity results measured by PSNR, SSIM and LPIPS demonstrate that our method generates videos more similar to the dense model compared with SVG under the same sparsity. Specifically, for Wan2.1 (768p), our method achieves non-marginal improvements over SVG on PSNR, SSIM and LPIPS (such as our 8.71 LPIPS *v.s.* 17.57 LPIPS from SVG under 75% sparsity). For the Hunyuan model, our method achieves better performance across almost all reported metrics, under a fair comparison with SVG following the same sparsity and computational cost in PFLOPs. We further provide the results with FP8 model in Table 2 of Appendix C, and our method maintains the advantages compared to the SVG method. We exclude additional overhead of spatial or temporal head selection when reporting PFLOPSs for SVG in Table 1. Note that the additional overhead of our DraftAttention is minor as shown in Table 3 of Appendix C.

**Superior Inference Acceleration.** Furthermore, we provide our latency results in Figure 4. The latency results are tested on H100 and A100 for both Huyuan and Wan2.1 models in 768p resolution. Our method achieves over 2× acceleration on an A100 GPU with 90% sparsity in the attention mechanism—demonstrating our outstanding practical efficiency.

**Better Visualization.** We provide the visualization for the comparison to SVG in Figure 5. All videos are generated with 90% sparsity. As highlighted in the red box, SVG exhibits a noticeable degradation in generation quality, with apparent blurry pixels. In contrast, our method better maintains the generation quality. We provide generated videos for further comparison in supplementary.

### 4.3 ABLATION STUDY

As shown in Figure 6, we provide the ablation study for the different kernels with average pooling and max pooling. The visualization is generated using 90% sparsity. The detailed results are included in Table 4 of Appendix C. We observe that average pooling achieves better generation quality. The ablation for different kernel size is included in Table 5 of Appendix C.

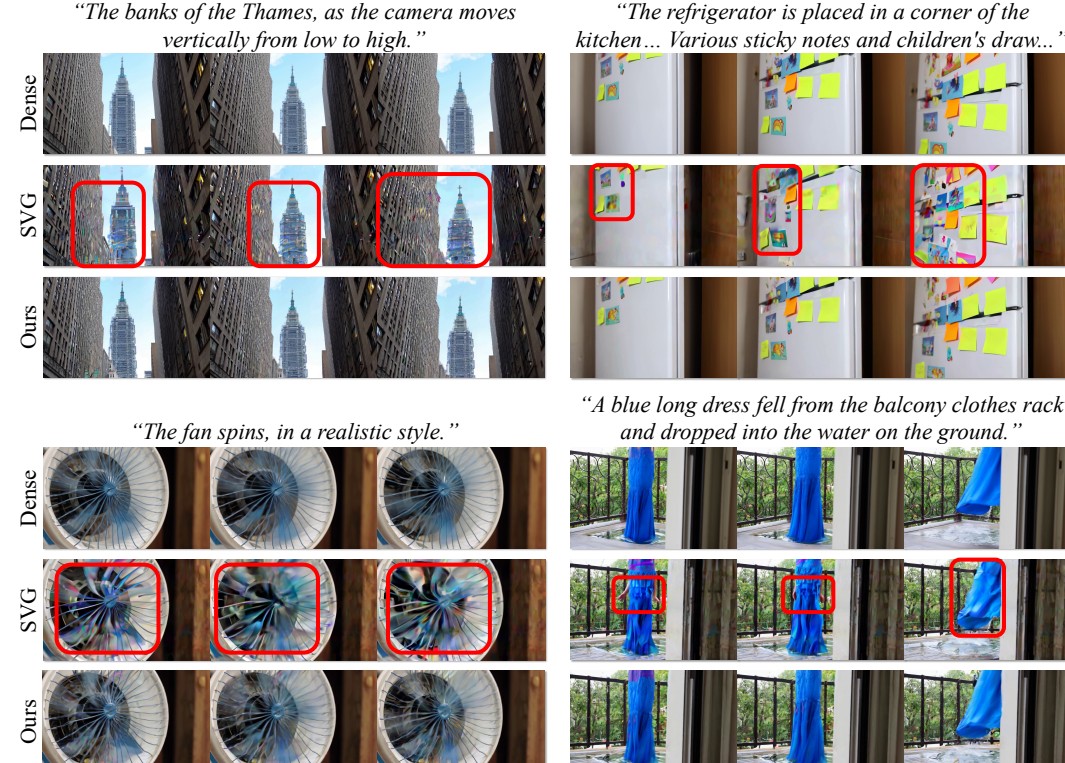

Figure 5: Visualization for our method and SVG (Xi et al., 2025) with 90% sparsity ratio in attention.

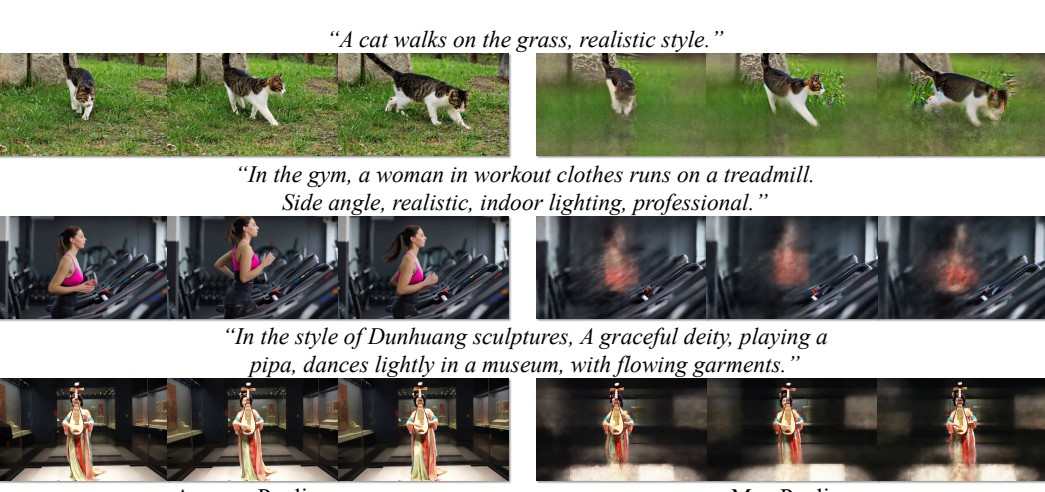

Figure 6: Visualization for the ablation study comparing average pooling and max pooling kernels.

## 5 CONCLUSION

In this paper, we propose **DraftAttention** for efficient video diffusion. We adopt pooling to compute a low-resolution draft attention map to guide the sparse attention over full-resolution query, key, and value representations. Combined with effective reordering, this approach achieves fast, hardware-friendly execution on GPUs. Theoretical analysis is further provided for the justification of our design. Experiments show that our method outperforms other methods and achieves up to $2\times$ end-to-end acceleration on GPUs. In the future work, we plan to introduce the quantization for the further acceleration of high-resolution and long-duration video generation on GPUs.

REPRODUCIBILITY STATEMENT

Our framework is primarily built upon 2D average pooling over latent video frames, which enables the identification of important spatial regions during video generation. The theoretical justification for the controlled difference between full-resolution attention and low-resolution draft attention is rigorously presented in the paper. All code and implementation details will be released publicly upon acceptance of the paper.

LLMs USAGE STATEMENT

LLMs were employed in a limited capacity to refine the organization and clarity of the manuscript narrative. All conceptual contributions and experiments were performed independently of LLMs.

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

APPENDIX

## A  DETAILED PROOF

### A.1  PROOF OF THEOREM 3.3

*Proof.* First, observe that for any $u \in R_i$ and $v \in R_j$, the draft attention assigns

$$(S_{\text{draft}})_{uv} = \widetilde{S}_{ij}, \quad \text{while} \quad S_{uv} = \langle Q_u, K_v \rangle. \tag{11}$$

By the definition of $\delta$, we have

$$|S_{uv} - (S_{\text{draft}})_{uv}| = |S_{uv} - \widetilde{S}_{ij}| \leq \delta. \tag{12}$$

Then, summing over all $n^2$ token pairs gives

$$\|S - S_{\text{draft}}\|_F^2 = \sum_{u,v} |S_{uv} - (S_{\text{draft}})_{uv}|^2 \leq n^2 \delta^2. \tag{13}$$

Taking square roots on both sides yields the desired result:

$$\|S - S_{\text{draft}}\|_F \leq \delta n. \tag{14}$$

This completes the proof. $\qquad\square$

### A.2  PROOF OF THEOREM 3.5

*Proof.* Fix a query row $u$. Let $\mathcal{D}(u) := \{v : \widehat{M}_{uv} = -\infty\}$ be the set of dropped indices, and denote by

$$\mu_u := \sum_{v \in \mathcal{D}(u)} P_{uv}$$

the probability mass that dense attention $P$ allocates to those entries.

*Step 1 (exact row deviation).* Write $Z_u = \sum_v e^{S_{uv}}$, $Z_u^{\text{K}} = \sum_{v \notin \mathcal{D}(u)} e^{S_{uv}}$, and $Z_u^{\text{D}} = \sum_{v \in \mathcal{D}(u)} e^{S_{uv}}$. Then $\mu_u = Z_u^{\text{D}}/Z_u$ and $Z_u^{\text{K}} = Z_u(1 - \mu_u)$. For $v \notin \mathcal{D}(u)$,

$$P_{uv} = \frac{e^{S_{uv}}}{Z_u}, \qquad P_{uv}^{\widehat{M}} = \frac{e^{S_{uv}}}{Z_u^{\text{K}}} = \frac{P_{uv}}{1 - \mu_u},$$

while for $v \in \mathcal{D}(u)$ we have $P_{uv}^{\widehat{M}} = 0$. Hence the absolute change on dropped indices sums to $\mu_u$, and on kept indices sums to

$$\sum_{v \notin \mathcal{D}(u)} \left| P_{uv} - \frac{P_{uv}}{1-\mu_u} \right| = \frac{\mu_u}{1 - \mu_u} \sum_{v \notin \mathcal{D}(u)} P_{uv} = \mu_u.$$

Therefore

$$\|P_{u,\cdot} - P_{u,\cdot}^{\widehat{M}}\|_1 = 2\mu_u.$$

Since $\|x\|_2 \leq \|x\|_1$ for each row difference vector, summing over rows gives

$$\|P - P^{\widehat{M}}\|_F \leq 2\sqrt{n} \max_u \mu_u.$$

*Step 2 (bounding $\mu_u$ by $\delta$ and keep ratio).* If a block $(i, j)$ is dropped, then $\widetilde{S}_{ij} < t$. By the deviation bound $|S_{uv} - \widetilde{S}_{ij}| \leq \delta$, all logits in this block satisfy $S_{uv} \leq t + \delta$. If a block is kept, then $\widetilde{S}_{ij} \geq t$, so $S_{uv} \geq t - \delta$. Let $N_D(u)$ and $N_K(u)$ be the numbers of dropped and kept tokens in row $u$. Then

$$Z_u^{\text{D}} \leq N_D(u)\, e^{t+\delta}, \qquad Z_u^{\text{K}} \geq N_K(u)\, e^{t-\delta}.$$

Therefore

$$\mu_u = \frac{Z_u^{\text{D}}}{Z_u^{\text{D}} + Z_u^{\text{K}}} \leq \frac{1}{1 + \frac{N_K(u)}{N_D(u)} e^{-2\delta}}.$$

Define $s_u := N_K(u)/(N_K(u) + N_D(u))$, the fraction of tokens retained in row $u$, and let $s := \inf_u s_u$. Then

$$\mu_u \leq \frac{1}{1 + \frac{s}{1-s} e^{-2\delta}}.$$

*Step 3 (final bound).* Substituting this into the Frobenius estimate from Step 1 yields

$$\|P - P^{\widehat{M}}\|_F \leq \frac{2\sqrt{n}}{1 + \frac{s}{1-s} e^{-2\delta}},$$

which is the claimed bound. □

## B  EXPERIMENTAL SETTINGS

**Different Codebases for Wan2.1.** Our method and SVG adopt different implementations/codebases for Wan2.1. Our baseline results (0% sparsity) for Wan2.1 strictly follow the original Wan2.1 codebase and settings to ensure consistency with the official model. In contrast, the SVG results for Wan2.1 are obtained from the codebase (Xi et al., 2025), which incorporates several modifications—such as the transformation of negative prompts to improve dynamic degree scores—resulting in different perceptual performance even for the same dense model at 0% sparsity.

Thus, it may not be entirely fair to directly compare the perceptual metrics across methods using Wan2.1, as the results are obtained with different codebases and experimental settings. In Table 1, the similarity results for Wan2.1 are obtained using the same implementations and thus their comparisons are fair. However, the perceptual metrics are primarily intended to compare the dense and sparse generation results within each method, rather than to serve as a cross-method benchmark.

For Hunyuan model, SVG does not change the codebase, and we share the same settings. As shown in Table 1, our method demonstrates better perceptual metrics than SVG. Furthermore, our method shows non-marginal improvements on similarity metrics over SVG for both Wan2.1 and Hunyuan models. The superior performance shows the effectiveness of our method.

**Prompts for Video Generation.**  In our experiments, we follow the protocol established in SVG (Xi et al., 2025), using prompts from the Penguin Video Benchmark for video generation and evaluating quality using the VBench metrics, which has become a common practice in recent video diffusion works Xi et al. (2025); Yang et al. (2025). This also ensures a fair and consistent comparison with prior work SVG.

**Sparsity Ratio.**  Our DraftAttention is only applied to part of diffusion steps. Following prior works Xi et al. (2025); Li et al. (2024a;b); Liu et al. (2024), we retain full attention for the first 25% of denoising steps to preserve the video generation quality. Furthermore, the model has other modules besides attention. Thus, even if our attention sparsity reaches 90%, it does not mean that the overall sparsity ratio for the whole model is 90%. Our 90% sparsity only means the sparsity of certain attention modules and steps.

## C  ADDITIONAL RESULTS

### C.1  RESULTS WITH FP8 QUANTIZATION

We further provide the results with FP8 model from Hunyuan Video in Table 2. We observe that, with FP8 model weights, our method maintains the advantages compared to the SVG method. Meanwhile, we further achieve up to **1.83**× acceleration with FP8 model weights in 90% sparsity on H100 GPU.

### C.2  OVERHEAD PROFILING

We provide the overhead profiling with 90% sparsity ratio with Hunyuan model in 768p resolution. According to the results in Table 3, we demonstrate that the latency overhead brought by the DraftAttention (including reorder) is minor in the table below, which is about 0.5% of the overall latency.

Table 2: Main results of our method compared to the Sparse VideoGen (SVG) Xi et al. (2025).

| Model | Method | Sparse Ratio | PSNR ↑ | SSIM ↑ | LPIPS ↓ | Img. Qual. | Sub. Cons. | Bakg. Cons. | Dyn. Deg. | Aes. Qual. | PFLOPs ↓ |
|---|---|---|---|---|---|---|---|---|---|---|---|
| Hunyuan (768p, fp8) | Dense | 0% | / | / | / | 65.9% | 96.1% | 97.2% | 34.4% | 58.6% | 682.67 |
| | SVG | 90% | 25.59 | 82.98 | 18.13 | 64.6% | 95.6% | 96.8% | 30.2% | 57.7% | 283.20 |
| | Ours | 90% | **26.77** | **84.37** | **13.93** | **65.5%** | **95.8%** | **97.0%** | **33.4%** | **57.9%** | 283.20 |

Table 3: Latency results under different sparsity levels and `DraftAttention` settings.

| Sparsity | 0% | 90% | 90% |
|---|---|---|---|
| DraftAttention | No | No | Yes |
| Latency (s) | 1947 | 1107 | 1113 |

## C.3 ABLATION FOR DIFFERENT POOLING KERNELS

We further provide the quantitive results with Hunyuan model in 768p and 90% sparsity in Table 4, our average pooling demonstrates superior generation performance than max pooling.

Table 4: Comparison of pooling strategies. Avg Pool achieves better image quality and subjective consistency across all metrics.

| Pooling Type | PSNR ↑ | SSIM ↑ | LPIPS ↓ | Img. Qual. | Sub. Cons. |
|---|---|---|---|---|---|
| Avg Pool | 24.22 | 79.90 | 18.12 | 65.9% | 95.7% |
| Max Pool | 16.31 | 63.29 | 25.81 | 54.2% | 91.4% |

## C.4 ABLATION FOR DIFFERENT POOLING KERNEL SIZE

We choose the kernel size $8\times16$, as the corresponding latent sizes—$32\times48$ for 512p and $48\times80$ for 768p—are perfectly divisible by the $8\times16$ kernel, thus ensuring efficient and artifact-free pooling during the practical diffusion. We provide the experimental results with additional ablation study (with Hunyuan model in 768p and 90% sparsity) on pooling kernel sizes demonstrated in Table 5. $8\times16$ leads to better performance than 16 16.

Table 5: Comparison of different kernel sizes.

| Kernel Size | PSNR ↑ | SSIM ↑ | LPIPS ↓ | Img. Qual. | Sub. Cons. |
|---|---|---|---|---|---|
| $8\times16$ | 24.22 | 79.90 | 18.12 | 65.9% | 95.7% |
| $16\times16$ | 23.81 | 78.19 | 20.06 | 65.4% | 95.3% |

