# OpenReview forum: "DraftAttention: Fast Video Diffusion via Low-Resolution Attention Guidance"
_ICLR.cc/2026/Conference — ICLR 2026 Conference Withdrawn Submission_

### Official Review · Reviewer_FEKX · 2025-10-17

**Soundness:** 2
**Presentation:** 3
**Contribution:** 1
**Rating:** 2
**Confidence:** 5

**Summary:**

The paper proposes DraftAttention, a training-free acceleration method for video diffusion transformers. It computes a low-resolution (pooled) draft attention over downsampled Q/K to select top-r region-to-region interactions, then applies a corresponding structured block-sparse mask at full resolution. A deterministic locality-preserving permutation (patch grouping) makes the masked pattern hardware-friendly. The authors give upper bounds on approximation error (pooling → draft vs. dense; sparsification vs. dense), and report up to ~2× E2E speedups at similar quality to dense or to a strong static baseline, with small overhead for reordering.

**Strengths:**

1. Simple and effective idea. Compatible with FP8.

2. Hardware-aware: the permutation bridges region-level masks to fixed-size block kernels.

3. Some theory with Frobenius-norm bounds.

4. Consistent quality at meaningful speedups; overhead appears small.

**Weaknesses:**

1. Missing highly relevant prior work.

a. Low-res (mean-pooled) attention as a coarse proxy. MoBA and Quest (Query-aware sparsity) are, to my knowledge, the earliest to explicitly use mean-pooling as coarse scoring in LLM (what this paper's authors refer to as  “low-resolution attention”); SpargeAttention and VSA later adopt analogous ideas in image/video.

b. Locality-preserving permutations. Sliding Tile Attention first proposes locality-preserving permutations for efficient block-sparse attention in video; SpargeAttention concurrently uses a Hilbert-curve variant. This paper's reordering techniques appears functionally identical.

**I find the lack of citation to highly relevant prior works in this paper deeply concerning**. This paper should (i) credit those works, (ii) make the novelty claim precise and (iii) clarify differences.

Lu, Enze et al. MoBA: Mixture of Block Attention for Long-Context LLMs. arXiv preprint arXiv:2502.13189 (2025).

Tang, Jiaming et al. QUEST: Query-Aware Sparsity for Efficient Long-Context LLM Inference. In ICML 2024.

Zhang, Peiyuan et al. STA: Fast Video Generation with Sliding Tile Attention. In ICML 2025.

Zhang, Jintao et al. SpargeAttention: Accurate and Training-free Sparse Attention Accelerating Any Model Inference. In ICML 2025.

Zhang, Peiyuan et al. VSA: Faster Video Diffusion with Trainable Sparse Attention. In Neurips 2025.


2.  It is a bit tricky to say the the attention sparsity is 90% because the first 25% timestep if kept dense attention. I would presume it is only 90% sparse for the later 75% timestep? It is a known fact the full denoising step (50) for video diffusio model is highly redundant. I wonder: a. how the model will perform if use use less dense attention step. b. What if we just reduce the total timestep from 50 to 25, and let the first 10 steps to be full attention. Will the human preference make a huge difference?

**Questions:**

1. The method fixes 2D pooling 8×16. Why not try 3D pooling to exploit temporal redundancy as well？

2. Implementation is less efficient on H100. I also wonder if the full attention baseline on H100 use FlashAttention2 or FlashAttention3.

---

### Official Review · Reviewer_baso · 2025-10-29

**Soundness:** 2
**Presentation:** 2
**Contribution:** 1
**Rating:** 2
**Confidence:** 5

**Summary:**

This paper proposes DraftAttention, a training-free framework to speed up video diffusion transformers while keeping good generation quality. It first computes a lightweight low-resolution "draft attention" (via downsampling query and key) to identify important regions and create a sparse mask. It also reorders tokens to make each region’s tokens contiguous. Experiments show it achieves good speedup on GPUs.

**Strengths:**

The paper presents DraftAttention, a training-free, plug-and-play sparse attention framework that can accelerate video diffusion transformers while maintaining generation quality.

The method is simple, generalizable, and practical for integration into existing models without retraining.

**Weaknesses:**

**Limited novelty**
  The paper mainly proposes two techniques, but both with limited novelty:
  **(a)** Mean pooling compression to obtain a low-resolution attention map for guiding sparse attention has appeared in SpargeAttn, SeerAttention, etc.
  **(b)** Token reordering to exploit token sparsity and improve hardware efficiency has also been studied in SpargeAttn and SVG2.
    Moreover, the proposed token permutation is simple (it can be implemented with one line of code in Eniops) and mainly serves to meet kernel memory contiguity requirements. It should not be presented as a key innovation.

**Incomplete experimental evaluation**
  **(a)** No comparison of attention kernel speed with baseline kernels.
  **(b)** No analysis of the relationship between sparsity ratio and kernel speed, which is crucial to assess kernel optimization.
  **(c)** No end-to-end model speed comparison with baselines.
  **(d)** The set of baselines is too limited: only *SVG* is included. Additional comparisons with *SVG2*, *SpargeAttn*, and *RadialAttn*, etc. are needed.
  **(e)** The kernel is implemented based on Block Sparse Attention from *FlashAttention 2*, which does not align with *FlashAttention 3* (the standard on H100 GPUs), weakening the paper’s practical contribution.
  **(f)** There is no kernel-level speed or overhead analysis, including the cost of token permutation and draft computation.

**Questions:**

What advantages does DraftAttention offer over SpargeAttn, which also employs mean pooling and token permutation, along with additional improvements such as sparse online softmax and selectively mean pooling? Since SpargeAttn is a relatively early but closely related work, why is there no direct comparison with it?

It would be better to discuss sparsity across different attention heads and layers, and to explain how to determine the sparsity ratio *r*. While there is a theoretical analysis of the error, there is no empirical verification of the error.

---

### Official Review · Reviewer_PbAn · 2025-10-30

**Soundness:** 3
**Presentation:** 3
**Contribution:** 2
**Rating:** 4
**Confidence:** 3

**Summary:**

The paper presents DraftAttention, a training-free method to speed up video diffusion transformers. It computes a low-resolution attention map to identify important regions and uses this to guide sparse attention at full resolution. Combined with a token reordering step for GPU efficiency, the approach achieves up to 2x faster generation with minimal quality loss compared to dense attention and outperforms prior sparse attention methods under the same compute budget.

**Strengths:**

1) The paper is well written; it is easy to read and understand.
2) The method is training-free and plug-and-play, it works with existing state-of-the-art video diffusion transformers.
3) Good empirical results: the approach achieves up to 2× speedup on A100/H100 GPUs while preserving video quality.
4) The method is orthogonal to other acceleration techniques; it can be combined with quantization or distillation for further efficiency gains.

**Weaknesses:**

1) The experiments mainly compare against Sparse VideoGen (SVG). Other sparse attention methods like AdaSpa are discussed but not included, which weakens claims of state-of-the-art performance.
2) The paper relies entirely on automated metrics (PSNR, SSIM, LPIPS, VBench) to assess perceptual quality. These metrics often fail to capture nuanced aspects of realism and user preference. A human study or preference test would significantly strengthen the quality claims.
3) There are no comparisons to other transformer acceleration techniques, like token merging, token pruning, feature caching across timesteps, and so on.

**Questions:**

1) Could you please provide VBench total score for entries in Table 1?
2) Currently, kernel for draft attention is only spatial. Have you also tried or considered 3D kernels (e.g., 4x8x16)? Would this method perform well in this scenario?
3) I’m curious whether the proposed method is compatible with step-distilled models such as DMD2, which can generate high-quality videos in just 1–4 sampling steps. If it only works with the original, non-distilled models, its practical utility becomes limited, since the method is not applied during the first 25% of diffusion steps, already more computationally expensive than the entire budget of these distilled models.

---

### Official Review · Reviewer_CeWm · 2025-10-31

**Soundness:** 3
**Presentation:** 3
**Contribution:** 2
**Rating:** 2
**Confidence:** 4

**Summary:**

This paper presents DraftAttention, a training-free acceleration framework for video diffusion transformers that introduces dynamic sparse attention guided by a low-resolution draft attention map. The approach computes attention at a downsampled resolution to identify spatially and temporally redundant regions, and then applies a block sparse mask to full-resolution attention computations. The method achieves up to 2× speedup in video synthesis with minimal loss in generation quality, outperforming existing sparse attention baselines according to the reported results.

**Strengths:**

- The presentation is clear and well organized, with thorough motivation, method description, and experimental validation.

- The computational bottlenecks in attention of video diffusion transformers are a pressing issue, and the proposed approach targets a key challenge for scaling video generation systems.

**Weaknesses:**

- The core idea—using pooling-based  approximations to guide sparse block attention—is conceptually similar to prior work (e.g., MInference for LLMs and SpargeAttention for video diffusion). This overlap weakens the novelty claim.

- The experimental evaluation omits direct comparisons with several relevant sparse or spatially adaptive attention methods such as SpargeAttention, SlidingTileAttention, RadialAttention, and XAttention, which are necessary to situate the method’s performance within the broader literature.

**Questions:**

- How does DraftAttention differ fundamentally from existing pooling-based block-sparse acceleration techniques (e.g., MInference)?

---

### Note · Authors · 2025-11-12

I have read and agree with the venue's withdrawal policy on behalf of myself and my co-authors.